# Type 2 Diabetes Induces a Pro-Oxidative Environment in Rat Epididymis by Disrupting SIRT1/PGC-1α/SIRT3 Pathway

**DOI:** 10.3390/ijms23168912

**Published:** 2022-08-10

**Authors:** Antónia Diniz, Marco G. Alves, Emanuel Candeias, Ana I. Duarte, Paula I. Moreira, Branca M. Silva, Pedro F. Oliveira, Luís Rato

**Affiliations:** 1CICS-UBI-Health Sciences Research Centre, University of Beira Interior, 6201-506 Covilhã, Portugal; 2Laboratory of Cell Biology, Unit for Multidisciplinary Research in Biomedicine (UMIB), Department of Microscopy, Institute of Biomedical Sciences Abel Salazar (ICBAS), University of Porto, 4500-313 Porto, Portugal; 3Laboratory for Integrative and Translational Research in Population Health (ITR), University of Porto, 4050-600 Porto, Portugal; 4Biotechnology of Animal and Human Reproduction (TechnoSperm), Institute of Food and Agricultural Technology, University of Girona, 17003 Girona, Spain; 5Unit of Cell Biology, Department of Biology, Faculty of Sciences, University of Girona, 17003 Girona, Spain; 6CNC-Center for Neuroscience and Cell Biology, Rua Larga, Faculty of Medicine (Pólo 1, 1st Floor), University of Coimbra, 3004-517 Coimbra, Portugal; 7Mitochondrial Toxicology & Experimental Therapeutics Laboratory, CNC-Center for Neuroscience and Cell Biology, UC-Biotech Building, Lot 8A, Biocant Park, 3060-197 Cantanhede, Portugal; 8Institute for Interdisciplinary Research (IIIUC), University of Coimbra, Casa Costa Alemão-Pólo 3, Rua D. Francisco de Lemos, 3030-789 Coimbra, Portugal; 9CIBB-Center for Innovative Biomedicine and Biotechnology, Rua Larga, Faculty of Medicine (Pólo 1, 1st Floor), University of Coimbra, 3004-504 Coimbra, Portugal; 10Institute of Physiology, Faculty of Medicine, University of Coimbra, 3000-548 Coimbra, Portugal; 11Faculdade de Ciências da Saúde, University of Beira Interior, Rua Marquês d’Ávila e Bolama, 6201-001 Covilhã, Portugal; 12LAQV-REQUIMTE and Department of Chemistry, University of Aveiro, 3810-193 Aveiro, Portugal; 13Health School of the Polytechnic Institute of Guarda, 6300-035 Guarda, Portugal

**Keywords:** type 2 diabetes mellitus, male infertility, epididymis, mitochondrial function, oxidative parameters

## Abstract

Diabetes mellitus type 2 (T2DM) has been associated with alterations in the male reproductive tract, especially in the epididymis. Although it is known that T2DM alters epididymal physiology, disturbing mitochondrial function and favoring oxidative stress, the mechanisms remain unknown. Sirtuin 1 (SIRT1), peroxisome proliferators-activated receptor γ coactivator 1α (PGC-1α), and sirtuin 3 (SIRT3) are key regulators of mitochondrial function and inducers of antioxidant defenses. In this study, we hypothesized that the epididymal SIRT1/PGC-1α/SIRT3 axis mediates T2DM-induced epididymis dysfunction by controlling the oxidative profile. Using 7 Goto-Kakizaki (GK) rats (a non-obese model that spontaneously develops T2DM early in life), and 7 age-matched Wistar control rats, we evaluated the protein levels of SIRT1, PGC-1α, and SIRT3, as well as the expression of mitochondrial respiratory complexes. The activities of epididymal glutathione peroxidase (GPx), glutathione reductase (GR), superoxide dismutase (SOD), and catalase (CAT) were determined, as well as the epididymal antioxidant capacity. We also evaluated protein nitration, carbonylation, and lipid peroxidation in the epididymis. The T2DM rats presented with hyperglycemia and glucose intolerance. Epididymal levels of SIRT1, PGC-1α, and SIRT3 were decreased, as well as the expression of the mitochondrial complexes II, III, and V, in the T2DM rats. We found a significant decrease in the activities of SOD, CAT, and GPx, consistent with the lower antioxidant capacity and higher protein nitration and lipid peroxidation detected in the epididymis of the T2DM rats. In sum, T2DM disrupted the epididymal SIRT1/PGC-1α/SIRT3 pathway, which is associated with a compromised mitochondrial function. This resulted in a decline of the antioxidant defenses and an increased oxidative damage in that tissue, which may be responsible for the impaired male reproductive function observed in diabetic men.

## 1. Introduction

Type 2 diabetes mellitus (T2DM) is a pandemic disease; it is estimated that, by the year 2030, more than 600 million people may suffer from this disease [1,2]. The incidence of T2DM has increased among young men and, currently, more than half of men with diabetes are subfertile/infertile [3]. Diabetes has several deleterious effects on male reproductive health, ranging from the most remarkable effects, such as decreased libido and erectile dysfunction, to the “hidden” molecular signatures that culminate with sperm dysfunction. The resultant hyperglycemic status has been identified as one of the major causes of the decline in male reproductive health since the metabolic reprogramming occurring in the major male reproductive organs—testis [4] and epididymis [5]—leads to an exacerbated glycolytic flux, which is often associated with a prooxidant environment in these tissues [6,7]. This is particularly relevant in the epididymis, the organ responsible for establishing a unique environment for protecting spermatozoa and allowing their concentration, maturation, and storage in a quiescent state. Diabetes causes regression of the epididymis, leading to decreased weight in the caput, corpus, and caudal regions [8], and these structural changes are often preceded by cellular and molecular changes. Studies have reported that T2DM-induced oxidative stress (OS) in the epididymis contributes to increased sperm DNA fragmentation [9]. In fact, high levels of 8-hydroxydeoxyguanosine in the semen of diabetic men were correlated with the levels of nitrite/nitrate in the seminal plasma [10]. Damages during sperm epididymal transit compromise downstream processes, such as sperm maturation and capacitation. This association has been supported by the decline of sperm quality and fertilization capacity in diabetic animals described by Kim and Moley [11]. Spermatozoa are particularly susceptible to oxidative injuries due to the lipidic composition of the plasma membrane, which is particularly rich in polyunsaturated fatty acids, and to the reduced intracellular antioxidant defenses. So, during maturation and storage, they rely on the epididymis to ensure the maintenance of OS levels. Rato and collaborators [7] have shown that DM decreases male fertility potential, promoting several alterations in testicular mitochondrial bioenergetics. Metabolic alterations were also observed in the epididymis of prediabetic rats [5]. Mitochondria are usually called the “cellular powerhouses” due to their role in the production of adenosine triphosphate (ATP), but they are also the main source of reactive oxygen species (ROS). On one hand, this organelle is extremely important to ensure normal spermatogenesis and, thus, for functional sperm; on the other hand, mitochondrial dysfunction is associated with male infertility [12]. Peroxisome proliferator-activated receptor gamma coactivator 1-α (PGC-1α) acts as a transcriptional regulator, playing an important role in the control of the expression of genes involved in energy homeostasis [13] and mitochondrial biogenesis [7]. PGC-1α stimulates the expression of transcriptional regulators, the nuclear respiratory factors 1 and 2 that act on the nuclear genes coding for subunits of the oxidative phosphorylation (OXPHOS) system [13]. Under conditions of excessive oxidative stress, PGC-1α is activated upstream by the silent information regulator 1 (SIRT1), a nicotinamide adenine dinucleotide (NAD^+^)-dependent deacetylase. SIRT1 deacetylates multiple residues in PGC-1α, promoting mitochondrial function in response to energy depletion [14]. SIRT1 is a member of the conserved family of proteins, the sirtuins (SIRT 1–7), belonging to the class III NAD^+^-dependent histone deacetylases. SIRT1 plays several physiological roles, including regulation of glucose metabolism, cell survival, and mitochondrial respiration. Knockdown of SIRT1 attenuates spermatogenesis, thus confirming the importance of this sirtuin in spermatogenesis and germ cell survival [15,16]. Previous studies have highlighted the relevance of the SIRT1/PGC-1α pathway in the maintenance of testicular function of diabetic animals [15], and the depletion of SIRT1 and PGC-1α induced by diabetes lead to germ cells loss [15]. After being activated, PGC-1α induces the expression of sirtuin 3 (SIRT3) [17]. SIRT3 is the most important deacetylase involved in the mitochondrial energy homeostasis. SIRT3 interacts with the enzymatic complexes of the electron transport chain (ETC), resulting in increased activity of complexes and contributing to an efficient electron flow [18]. PGC-1α and SIRT3 act in synergy, not only to support mitochondrial function, but also to induce the antioxidant defense system [17]. Our group described, for the first time, that prediabetes reduces the testicular levels PGC-1α and SIRT3, and such a decrease is associated with increased testicular OS [7] and declined reproductive parameters [6]. The available evidence has allowed us to understand in detail how testicular bioenergetics are controlled, but the role of the molecular axis SIRT1/PGC-1α/SIRT3 at the level of the epididymis remains uncertain. Since these proteins are interconnected, disruption of the pathway SIRT1/PGC-1α/SIRT3 may be the molecular basis of the mitochondrial dysfunction prompting a prooxidant environment.

Thus, we aimed to study the effects of T2DM on the molecular pathways underlying the control of epididymal function, focusing on the SIRT1/PGC-1α/SIRT3 pathways. We also evaluated the effects of T2DM in the expression of mitochondrial respiratory complexes and measured the activities of the enzymes involved in the antioxidant defense system, as well as the epididymal antioxidant capacity, protein carbonylation, protein nitration, and lipid peroxidation.

## 2. Results

### 2.1. General Characteristics of the T2DM Animal Model

T2DM GK rats are a non-obese model and spontaneously develop T2DM [19]. These animals presented with a significant decrease (by 11%) in body weight (418.80 ± 6.11 g) when compared to the control rats (470.60 ± 11.95 g; statistical power (SP) = 100%) (Table 1). At the end of the treatment, glycemic values of the T2DM rats were significantly increased (342.50 ± 42.08 mg/dL) when compared to the control group, which showed 90.63 ± 6.70 mg/dL (SP = 100%) (Table 1). Blood HbA1c levels were also significantly increased (by 85%) in the T2DM group (8.29 ± 0.19%) when compared to the control animals (4.46 ± 0.07%) (SP = 100%) (Table 1). These results denote a prolonged state of hyperglycemia and impaired glucose metabolism. In fact, the results attained for the glucose tolerance test show that blood glycemia of the T2DM group increased during the 120 min of the test (Figure 1a), indicating the development of glucose intolerance. This can be seen by the significant increase (by 137.1%) of the AUC_GTT_ values in the T2DM animals (47,744 ± 2829 arbitrary units (a.u.)) when compared to animals from the control group (20,010 ± 1123 a.u.) (SP = 100%) (Figure 1b). Then, we evaluated the insulin status and observed a significant increase in the fasting insulin of the T2DM group (0.26 ± 0.02 ng/mL), whereas the control group only exhibited (0.15 ± 0.02 ng/mL) (SP = 100%) (Table 1). The higher levels of insulin were consistent with the increased HOMA-IR in the T2DM animals (3.58 ± 0.36) as compared to the control animals, which showed (0.67 ± 0.07), thus corroborating that the T2DM animals developed insulin resistance (SP = 100%) (Table 1). Furthermore, we also observed a significant decrease (by 25%) in the gonads’ weight of the T2DM rats (2.81 ± 0.03 g), as compared to that of the rats from the control group (3.77 ± 0.12 g) (SP = 100%) (Table 1), accompanied by a decrease in the gonadosomatic index in the T2DM group (0.66 ± 0.02) when compared to the control rats (0.80 ± 0.03) (Table 1). Concerning the epididymis, those of the T2DM group presented with a weight of 1.08 ± 0.02 g, whereas as those of the animals of the control group had a weight of 1.47± 0.05 g, corresponding to a 27% decrease (SP = 100%) (Table 1).

### 2.2. T2DM Compromised the Expression of Key Regulators of Biogenesis in the Epididymis 

We found a significant decrease (by 22%) in the expression of the SIRT1 in the epididymal tissue of the T2DM group animals (0.78 ± 0.12-fold variation to control) when compared with control group rats (1.00 ± 0.09-fold variation) (SP = 86%) (Figure 2). These results led us to assess the protein levels of PGC-1α in the epididymis, and we observed that the rats in the T2DM group showed lower levels of PGC-1α (0.68 ± 0.22-fold variation to control) (by 32%) in relation those of the control group (1.00 ± 0.09-fold variation) (SP = 96%) (Figure 2). Then, we quantified the expression of SIRT3 and observed that T2DM induced a significant decrease in the expression of this deacetylase in the epididymal tissue. The rats of the T2DM group exhibited a decrease in the expression levels of SIRT3 (0.65 ± 0.11-fold variation to control), corresponding to a 35% reduction when compared to the animals of the control group (SP = 93%) (Figure 2).

### 2.3. Expression of the Epididymal Mitochondrial Complexes II, III, and V Was Affected by T2DM

T2DM significantly decreased the expression of the mitochondrial complex II, from a 1.00 ± 0.12-fold variation in the epididymis of rats from the control group to a 0.76 ± 0.09-fold variation in that of the animals from the T2DM group, corresponding to 24% of the decrease (SP = 62%) (Figure 3). Also, we observed a decreased expression of mitochondrial complex III (by 22%) with the T2DM group (0.78 ± 0.10-fold variation to control) when compared to the control group (1.00 ± 0.12-fold variation) (SP = 62%). Complex V was decreased in the T2DM group by 23% (0.77 ± 0.15-fold variation to control) when compared to the control group (1.00 ± 0.12-fold variation). No differences were found concerning complex I and complex IV (SP = 63%) (Figure 3).

### 2.4. T2DM Significantly Decreases the Activity of Antioxidant Enzymes in the Epididymis

T2DM favors the increase of the oxidative environment in tissues due to the unbalanced production of ROS and an inefficient antioxidant system. We used a FRAP assay to evaluate the effects of T2DM in the epididymal antioxidant capacity. Our results showed a slight but significant alteration in the antioxidant capacity of the epididymal tissue of the T2DM animals (4.79 ± 0.12-fold variation to control) when compared to that of the control group (5.50 ± 0.32-fold variation to control) (SP = 62%) (Figure 4a). Our results showed that the activity of SOD was significantly decreased by 87% in the epididymal tissue of rats from the T2DM group (4628 ± 545.90 U/mg of protein), compared to the animals of the control group (36,535 ± 13,554 U/mg of protein) (SP = 100%) (Figure 4b). Similarly, CAT activity was decreased by 25% in the T2DM group animals (1991 ± 165.30 U/mg of protein), with the epididymis of animals from the control group exhibiting an activity of 2668 ± 192.80 U/mg of protein (SP = 90%) (Figure 4c). The activity of GPx was also significantly decreased by 58%, with the animals of the T2DM group exhibiting an epididymal activity of 4.93 ± 1.12 U/mg of protein and the control group an activity of 11.89 ± 0.74 U/mg of protein (Figure 4d). Contrastingly, we measured the activity of GR in the epididymal tissue, which was significantly increased by 30%, from 0.14 ± 0.03 U/mg of protein in the animals of the control group to 0.20 ± 0.02 U/mg of protein in those of the T2DM group (SP = 61%) (Figure 4e).

### 2.5. T2DM Increased Protein Nitration in the Epididymal Tissue

Proteins are targeted by ROS and reactive nitrogen species (RNS), suffering oxidation. So, we evaluated the impact of T2DM in epidydimal protein carbonylation and nitration by measuring the levels of DNP and 3-NT (3-nitrotyrosine), respectively. Animals from the T2DM group exhibited a decrease of 28% in the epididymal carbonyl content (0.60 ± 0.09-fold variation to control) when compared to the control group (0.84 ± 0.10-fold variation) (SP = 62%) (Figure 5a), while protein nitration was significantly increased by 48% in the epididymis of the T2DM animals (1.23 ± 0.11-fold variation to control) when compared to that of the control rats (0.83 ± 0.10) (SP = 86%) (Figure 5b). We also quantified the epididymal lipid peroxidation by measuring the relative levels of 4-HNE and observed a significant increase (by 40%) in the T2DM rats (1.22 ± 0.10-fold variation to control) in relation to those of the control group (0.87 ± 0.09) (SP = 84%) (Figure 5c).

## 3. Discussion

T2DM is one of the most prevalent metabolic diseases and is considered a public health concern. Increasing evidence suggests that T2DM has an adverse effect on male reproductive function. In this study, we used a T2DM model to evaluate the effects of T2DM in the epididymal SIRT1/PGC-1α/SIRT3 molecular axis, as well as in mitochondrial bioenergetics and OS. GK rats are characterized by being a non-obese model that spontaneously develop T2DM [19]. Four weeks after birth, the animals displayed several characteristics linked to T2DM and, in this work, after 8 months of age, animals presented with a stable, fasting hyperglycemia, without developing ketotic acidosis. At the end of the treatment, the hyperglycemia was pronounced in the animals from the T2DM group when compared to those of the control group. We also observed a significant glucose intolerance and increased levels of fasting insulin, illustrating insulin resistance, which was corroborated by the increased HOMA-IR. Concerning the reproductive function, T2DM animals presented with a significant decrease of the gonadosomatic index, as well as in the reproductive accessory organs, indicating that the structures of the testes and epididymis were affected by T2DM. This is consistent with other studies of diabetic models [20,21], and in diabetic men, showing declined sperm quality [22], with a decrease in sperm concentration, motility, and fertilization capacity, as well as subsequent embryo development [11]. Apart from morphologic alterations induced by T2DM, there are molecular changes that are based on male infertility related with T2DM. 

Indeed, several studies have linked the subtle metabolic alterations induced by the prodromal stage of DM, which are sufficient to induce metabolic alterations and affect the oxidative status in the testicular and epididymal tissues [5,23]. Mitochondria are responsible for the maintenance of essential events for cellular viability, playing an important role in the homeostasis of the redox environment. Thus, ensuring a normal mitochondrial function within the reproductive tract is imperative, particularly under DM, which induces severe fluctuations in energy levels. DM promotes alterations in the whole cellular metabolism, such as the unbalance of NAD^+^/NADH ratio [4] and, therefore, in sirtuin activity, since NADH is an essential co-substrate that directly regulates sirtuin activity [24]. T2DM adversely affects testicular metabolic pathways, particularly glucose metabolism [4] and fatty acid oxidation through a SIRT1/PGC-1α-dependent mechanism, leading to the accumulation of metabolic intermediates that cause testicular oxidative damage and germ cell apoptosis [15]. Compelling evidence highlights the relevance of SIRT1 in male fertility because SIRT1 knockout animals are infertile [25,26,27], showing decreased reproductive organ size, and are not able to successfully mate [25,28]. Sperm from SIRT1 knockout mice has abnormalities in morphology and in molecular composition [29]. SIRT1-deficient mice present with an arrest of the spermatogenic process due to dysfunctional mitochondria [26]. Moreover, epididymal sperm from SIRT1-mutant animals are frequently immature and much less motile than those from wild-type individuals, suggesting that maturation within the epididymis is also defective. In our study, we found a significant decrease in the epididymal SIRT1 protein levels in the T2DM animals. Although there are no data reporting the effects of T2DM in epididymal SIRT1 levels, our results are consistent with other models showing that the expression of SIRT1 is affected by DM [30]. SIRT1 interacts with mitochondrial proteins, and both in vitro and in vivo studies indicate that SIRT1 is a major regulator of PGC-1α [31,32]. Similar to SIRT1, the expression of PGC-1α in the epididymal tissue was significantly decreased, illustrating possible bioenergetic disruption in the epididymis of the T2DM animals. The SIRT1/PGC-1α axis is a key regulatory point for the maintenance of mitochondrial function and bioenergetic capacity since, together, both proteins are involved in “mitochondrial renewal” [33]. The decreased levels of PGC-1α might be related to the decrease in the SIRT3 expression observed in the epididymal tissue. As previously reported, PGC-1α is pivotal for the function of SIRT3 [17] and we have found that, even in the prodromal stages of T2DM, the expression of PGC-1α and SIRT3 is impaired [7]. Disruption of the SIRT1/PGC-1α/SIRT3 pathway has deleterious effects since the action of these proteins potentiates the expression and/or activity of several proteins and enzymes required for the proper functioning of mitochondria [7,31,32]. Sirtuins are considered a key target of pharmaceutical intervention to counteract diabetes-induced subfertility/infertility. Natural and synthetic products have been used to enhance sirtuin activity [34]. For instance, resveratrol, a natural compound present in the skin of grapes, blueberries, raspberries, mulberries, and in wine [35], is a potent SIRT1 activator [36]. Evidence has showed how this phenolic compound protects mice against male infertility associated with diet-induced obesity and insulin resistance [37]. The most pronounced effects of resveratrol were observed in mitochondrial activity, due to an improvement of antioxidant enzyme activities, such as CAT, GR, SOD [36]. The molecular basis between the positive effects of resveratrol or other similar compounds and male fertility may be mediated by sirtuins [37], since the structural characteristics of these proteins enable the interaction with compounds rich in polyphenols [38].

Next, we quantified the expression of mitochondrial respiratory complexes and found a significant decrease in the protein levels of complexes II, III, and V. This is in agreement with the fact that even a partial loss of PGC-1α and SIRT3 leads to a dysfunctional ETC [7]. Mitochondrial complex II has no contribution to the overall ETC process since no protons are released to the intermembrane space in this step of OXPHOS. The electrons flow sequentially through cytochrome C reductase (complex III), which funnels electrons from the coenzyme Q pool to cytochrome C. The decrease in the expression of mitochondrial complexes III and V may be due to the decreased levels of PGC-1α, since the expression of complex III and V encoded genes is PGC-1α-dependent [39]. Bearing in mind that complex III is the main source of ROS and complex V is responsible for ATP synthesis, the reduction of these two complexes will contribute to an inefficient ETC, leading to a possible production of oxygen-free radicals and low ATP levels [40]. Indeed, under diabetic conditions, the mitochondrial function is compromised by the impairment of ETC complex expression and/or activity. Under these conditions, Raza and collaborators [40] observed a reduction in mitochondrial complexes III and V, which has major implications in the maintenance of the oxidative environment. 

Epididymis plays an important role in the maturational process of sperm, and the microenvironment within this tubule ensures sperm viability. The normal functionality of the sperm requires a delicate balance between ROS production and recycling. Indeed, small quantities of ROS are required for sperm function, such as motility, but also for capacitation and acrosome reaction, which are pivotal for fertilization [41]. However, uncontrolled production of ROS is detrimental for sperm since the presence of polyunsaturated fatty acids (PUFAs) in plasma membranes is a preferred target for ROS. Spermatozoa are highly susceptible to oxidative injuries since they are practically devoid of ROS-scavenging enzymes and are dependent on the existing antioxidant protection in the male reproductive tract, particularly in the epididymis [42]. As previously mentioned, SIRT1 is a major upstream activator of PGC-1α which, in turn, acts in concert with SIRT3 to activate the ROS detoxifying pathway in a forkhead box O3-dependent mechanism [17]. We evaluated the activity of the main antioxidant enzymes of the reproductive tract, specifically SOD, CAT, GPx, and GR. We observed that T2DM significantly decreased the activity of SOD. SOD is one of the most important enzymes since it converts the superoxide anion radical into H_2_O_2_ and O_2_ [43]. In the epididymal tissue of diabetic animals, the activity of SOD is often reduced because there is an excessive production of ROS that SOD tries to counteract [44]. After being produced, H_2_O_2_ is converted by CAT into H_2_O and O_2_. Like SOD, the activity of CAT is also significantly decreased, indicating that T2DM inhibits severely the activities of the two main antioxidant enzymes within the reproductive male tract [44]. H_2_O_2_ can also be metabolized by GPx into H_2_O. In this work, the activity of GPx was significantly reduced in the epididymis of the T2DM animals, whereas GR presented a significant increase. This is in line with what was previously observed in the T2DM animal model and may be explained as an attempt to regenerate the levels of GPx, which might have been oxidized by a GPx-independent pathway [20]. In the array of mitochondrial defense mechanisms, GPx is one of the most important antioxidant enzymes. To detoxify ROS, GPx catalyzes the reduction of H_2_O_2_ into two H_2_O molecules using GSH as a hydrogen donor. After being oxidized, GSH is transformed into oxidized glutathione (GSSG), which is recycled by GR using NADPH into the initial form, GSH [45]. Indeed, the impairment of SIRT1/PGC-1α/SIRT3 in the epididymis of the T2DM rats may underlie the disruption of the antioxidant defense system, prompting an exacerbated production of ROS. These results were concordant with the decreased antioxidant capacity of the epididymal tissue. Inefficient antioxidant capacity is often associated with increased oxidative stress parameters [23]. The consequences of ROS may be evaluated through potential biomarkers of OS. In our work, we evaluated protein carbonylation and found that there are no alterations in protein carbonyl content. In addition, overproduction of ROS oxidizes PUFAs in cellular membranes through free radical chain reactions and forms lipid hydroperoxides as primary products [46], which may decompose and lead to the formation of reactive lipid electrophiles, especially 4-hydroxy-2-nonenal (4-HNE) [46]. Taking into account the huge amount of PUFAS in sperm membranes, we measured 4-HNE epididymal levels and observed a significant increase in epididymis lipid peroxidation of rats from the T2DM group. This is consistent with what was observed by Chatterjee and collaborators [44], who also found increased lipid peroxidation during maturation and storage in epididymis under diabetic conditions. Accordingly, high levels of lipid peroxidation were found in sperm with reduced motility, as well as in sperm mitochondrial membranes [47]. Several molecules are altered due to interactions with ROS, including reactive nitrogen species (RNS) in the microenvironment, and those changes in response to increased redox stress are considered biomarkers of OS, such as protein nitration. Proteins contain surface-exposed tyrosine (Tyr) residues in their composition, due to the hydrophobic nature of the amino acid, which allow the process of nitration. These proteins suffer post-translational modifications, mainly caused by the addiction of NO to the three positions of either free or protein-bound Tyr, resulting in the formation of 3-NT. We found an increase in 3-NT epididymal content, evidencing the susceptibility of epididymal tissue to RNS species, which may be explained, in part, by the possible dysfunction of ETC. Indeed, a recent study also showed that nitro-oxidative stress occurred in the head and in the midpiece of the sperm, which is linked with a dysfunctional ETC [48].

## 4. Materials and Methods

### 4.1. Chemicals

Anti-3-Nitrotyrosine antibody (9691S) and anti-SIRT3 (D22A3) were purchased from Cell Signaling (Danvers, MA, USA); rabbit-anti 4-Hydroxinonenal (ab5605) and total OXPHOS (ab110,413), was purchased from Abcam (Cambridge, UK); anti-DNP antibody (D9656) and anti-α-tubulin (T9026) were purchased from Sigma-Aldrich (St. Louis, MO, USA); rabbit anti-SIRT1 (sc-15404) and rabbit anti-PGC-1α (sc-13067) were purchased from Santa Cruz Biotechnology (Dallas, TX, USA); β-actin (MA515739) was purchased from Thermo Fischer (Waltham, MA, USA). The secondary antibodies goat anti-rabbit IgG-HRP (sc-2004), mouse anti-goat IgG-HRP (sc-2354), and goat anti-mouse IgG-HRP (sc-2005) were purchased from Santa Cruz Biotechnology (Dallas, TX, USA). WesternBright ECL substrate was purchased from Advansta (Menlo Park, CA, USA). The Genomic DNA Kit Tissue GK03.0100) was purchased from GRISP, Lda (Porto, Portugal). The iTaq^TM^ Universal SYBR Green Supermix and the Rat Insulin Enzyme Immunoassay kit were purchased from SPI-BIO, Bertin Pharma (Montigny le Bretonneux, France). All other chemicals were purchased from Sigma-Aldrich (St. Louis, MO, USA).

### 4.2. Animal Model and Experimental Design

The present study used 14 male rats: 7 middle-aged (8 months old) Goto-Kakizaki (GK) rats (a non-obese model that spontaneously develop T2DM early in life [49]), and 7 age-matched Wistar control rats. Figure 6 depicts the experimental layout of the work. The Wistar and GK rats were obtained from Charles River (Barcelona, Spain) and Taconic (Ejby, Denmark), respectively, and maintained in the animal colony of the Animal Research Center of the University of Coimbra, under controlled light (12 h day/night cycle) and humidity (45–65%), with standard hard pellet chow and sterile water *ad libitum*. Both groups received a sterile saline infusion. Signs of distress were carefully monitored and a glucose tolerance test (GTT) was used as a selection index. All animal experiments were conducted upon ethical approval by the Animal Welfare Committee of the Center for Neuroscience and Cell Biology and the Faculty of Medicine, University of Coimbra. All animal experiments were performed according to the “Guide for the Care and Use of Laboratory Animals” published by the US National Institutes of Health (NIH Publication No. 85–23, revised 1996), and the European Directives for the care and handling of laboratory animals (Directive 2010/63/EU). In accordance with Portuguese law (Ordinance no. 1005/92 of 23 October), the research team requested permission to perform this animal experimentation study from the Portuguese “Direcção Geral de Veterinária” (Portuguese Veterinarian and Food Department).

### 4.3. Glucose Tolerance Test

At 8 months of age, the rats of the control group and the T2DM rats were submitted to a glucose tolerance test, as described by Candeias and collaborators [49]. Briefly, approximately 16 h before the test, food was taken from the cages and animals were fasted. Their blood glucose levels were determined before intraperitoneal injection of 2 mg D-glucose/g body weight (basal glycemia), and after 15, 30, 60, and 120 min. At the end of the test, the cages were supplied with wet food. Results were expressed as milligrams of glucose per deciliter of blood and as the area under the curve (AUC).

### 4.4. Insulin Levels

Before the intraperitoneal injection of D-glucose (0 min), blood from the caudal vein of the fasted rats was collected and centrifuged at 572× *g* in a Sigma 2–16 PK centrifuge for 10 min at 4 °C. The resulting plasma was used to determine fasting insulin levels through the Rat Insulin Enzyme Immunoassay kit (Montigny le Bretonneux, France), according to the manufacturer’s instructions. Absorbance was read at 405 nm in a SpectraMax Plus 384 multiplate reader and, after maximum binding (B0), wells ranged from 0.2 to 0.8 arbitrary units. Results were expressed as nanograms per milliliter for plasma insulin levels. The homeostasis assessment model-insulin resistance (HOMA-IR) index was calculated using the formula: HOMA-IR = (fasting insulin [μU/mL] × fasting glucose [mmol/L])/22.5 [50].

### 4.5. Total Protein Extraction

The epididymal tissue was homogenized in an appropriate volume of lysis buffer (with freshly added 20 mM of sodium fluoride, 100 mM sodium orthovanadate, and 1% of protease inhibitor cocktail) and allowed to stand 20 min on ice. The homogenates were centrifuged at 14,000× *g* in the Hettich Mikro 200 R centrifuge for 20 min at 4 °C. After centrifugation, the pellet was discarded. The total protein concentration was quantified using the Bradford Protein Assay Kit II from Bio-Rad (Hercules, CA, USA) according to the manufacturer’s instructions, and the absorbance was measured by the xMark Microplate Spectrophotometer from Bio-Rad (Hercules, CA, USA). The protein concentration was determined using different bovine serum albumin (BSA) concentrations as standards for calibration. Optical densities of samples were determined at 595 nm [7].

### 4.6. Ferric Reducing Antioxidant Power

The ferric reducing antioxidant power (FRAP) assay was performed according to the colorimetric method described by Benzie and Strain [51]. Briefly, epididymal tissue was homogenized in a phosphate buffer (pH 7.4). Protein concentration was determined by the Bradford microassay, using BSA as a standard. The working FRAP reagent was prepared by mixing acetate buffer (300 mM, pH 3.6), 2,4,6-Tripyridyl-s-Triazine (TPTZ) (10 mM in 40 mM HCl), and FeCl_3_ (20 mM) in a 10:1:1 ratio (v:v:v). 180 µL of this reagent was mixed with 10 µg of the tissue homogenate. The reduction of the Fe^3+^–TPTZ complex to a colored Fe^2+^–TPTZ complex was monitored immediately after adding the sample, and 40 min later, by measuring the absorbance at 595 nm using a spectrophotometer (Bio-Rad xMARK^TM^, Microplate Spectrophotometer). The antioxidant potential of the samples was determined against standards of ascorbic acid, which were processed in the same manner as the samples. Absorbance results were corrected by using a blank, with H_2_O instead of the sample.

### 4.7. Analysis of Protein Nitration and Lipid Peroxidation

To evaluate the protein nitration and lipid peroxidation, slot-blot was used, as described previously [52]. Each sample of epididymal protein extract was diluted to a concentration of 0.05 µg/µL, using phosphate buffer saline (PBS), and transferred to a polyvinylidene difluoride (PVDF) membrane. The membranes were previously activated for 1 min in methanol, 5 min in sterile H_2_O, and 15 min in PBS. The technique was performed using a Hybrid-Slot manifold system (Biometra, Göttingen, Germany). The membranes were then blocked for 60 min with 5% non-fat milk Tris buffer solution containing 0.05% Tween 20 (TBS-T). Afterward, membranes were incubated overnight with rabbit anti-3-Nitrotyrosine antibody (1:5000; 9691S) and goat anti-4-Hydroxinonenal (1:5000; AB5605). Samples were visualized using a secondary antibody goat anti-rabbit IgG-HRP (1:10,000; sc-2004) and mouse anti-goat IgG-HRP (1:10,000; sc-2354), respectively. Membranes were then reacted with WesternBright™ ECL (Advansta, Menlo Park, CA, USA) and visualized on the Chemidoc MP Imaging System from Bio-Rad (Hercules, CA, USA). Densities from each band were obtained with Image Lab Software 5.1 from Bio-Rad (Hercules, CA, USA).

### 4.8. Analysis of Carbonyl Groups

Protein carbonyl groups were evaluated by slot-blot as described by [52]. Samples were derivatized using 2,4-dinitrophenylhydrazine, as described previously [53]. Briefly, 5 ug of lyophilized epididymal tissue homogenized in PBS was mixed with the same volume of sodium dodecyl sulfate (12%). The samples were then mixed with two volumes of 2,4-dinitrophenylhydrazine 20 mM diluted in trifluoroacetic acid 10% and incubated in the dark for 15–20 min at room temperature. Afterward, 1.5 volumes of 2 M Tris with 18% of β-mercaptoethanol was added to the samples to stop the reaction. Samples were then diluted to a concentration of 0.05 µg/µL using phosphate buffer saline (PBS). PVDF membranes were activated for 1 min in methanol, 5 min in sterile H_2_O, and 15 min in PBS. Slot-blot was performed using a Hybrid-Slot manifold system (Biometra, Göttingen, Germany). The membranes were then blocked for 60 min with 5% non-fat milk TBS-T, containing 5% skimmed dried milk. Then, the blocked membranes were incubated overnight with rabbit anti-DNP antibody (1:5000; D9656, Sigma-Aldrich). Membranes were then incubated using a secondary antibody goat anti-rabbit IgG-HRP (1:10,000; sc-2004, Santa Cruz Biotechnology, USA). Membranes were reacted with WesternBright™ ECL (Advansta, Menlo Park, CA, USA) and visualized on the Chemidoc MP Imaging System from Bio-Rad (Hercules, CA, USA). Densities from each band were obtained with Image Lab Software 5.1 from Bio-Rad (Hercules, CA, USA).

### 4.9. Western Blot

Western blot analysis was performed according to [54]. The total protein extracted from the epididymal (50 μg) was mixed briefly with the supplemented lysis buffer plus the loading buffer (50% Glycerol (*v*/*v*), 20% Tris-HCl (*v*/*v*), 10% Sodium dodecyl sulfate (SDS) (*w*/*v*), 1.25% β-mercaptoethanol (*v*/*v*), and 0.05% bromophenol blue (*v*/*v*), pH = 6.8). Samples were separated in 11% polyacrylamide gel (SDS-page). After electrophoresis, the proteins were transferred to previously activated PVDF membranes. Then, the membranes were blocked at room temperature with 5% non-fat milk TBS-T containing 5% skimmed dried milk. The membranes were then incubated overnight at 4 °C with rabbit polyclonal primary antibody against PGC-1α (1:1000, sc-13067 rabbit), SIRT3 (1:1000; D22A3) and rabbit anti-SIRT1 (1:1000, sc-15404), total OXPHOS (1:1000; ab110413), β-actin (1:10,000; MA515739), and anti-α-tubulin (1:10,000; T9026). After washing in TBS, the membranes were incubated with a secondary antibody goat anti-rabbit IgG-HRP (1:10,000; sc-2004) and goat anti-mouse IgG-HRP (1:10,000; sc-2005), respectively. Membranes were reacted with WesternBright™ ECL (Advansta, Menlo Park, CA, USA) and visualized with the Chemidoc MP Imaging System from Bio-Rad (Hercules, CA, USA). Densities from each band were obtained with Image Lab Software 5.1 from Bio-Rad (Hercules, CA, USA). The band density attained was divided by the corresponding α-tubulin or β-actin band intensity and expressed in fold variation versus the control group.

### 4.10. Enzymatic Assays

#### 4.10.1. Glutathione Peroxidase Activity

The quantification of Glutathione Peroxidase (GPx) activity was performed using an indirect determination assay, based on the oxidation of glutathione to oxidized glutathione (GSSG), catalyzed by GPx, which is then coupled to the recycling of GSSG back to glutathione, utilizing glutathione reductase (GR) and NADPH. The oxidation of NADPH to NADP^+^ is indicative of GPx activity. To perform this assay, we diluted 50 µg of the protein from the epididymal tissue in glutathione peroxidase assay buffer (50 mM of Tris-HCl, pH 8.0 containing 0.5 mM EDTA). Then, we mixed 5 mM of NADPH, 42 mM of reduced glutathione, and 10 U/mL of glutathione reductase with the assay buffer. The reaction was started by the addition of 30 mM butyl hydroperoxide solution. Glutathione peroxidase activity was measured following the decrease of absorbance at 340 nm using a spectrophotometer (UltrospecR 3000, Pharmacia Biotech, Cambridge, UK) at 25 °C. The activity of the enzyme was determined using the molar extinction coefficient of 6.22 mM^−1^·cm^−1^ and expressed as Units/mg.

#### 4.10.2. Glutathione Reductase Activity

The assay of Glutathione Reductase activity was based on the reduction of oxidized glutathione (GSSG) and posterior reconversion to reduced glutathione (GSH) by Glutathione Reductase, using one molecule of NADPH. Briefly, 100 µg of epidydimal protein homogenate was diluted in a glutathione reductase assay buffer (100 mM potassium phosphate buffer, pH 7.5, with 1 mM EDTA) and 1 mg/mL BSA. Then, this mixture was incubated for 10 min. at 25 °C in a reaction buffer containing 2 mM oxidized glutathione solution, the glutathione reductase assay buffer and 3 mM of 5,5-dithiobis (2-nitrobenzoic-acid). The reaction was started by the addition of the 2 mM NADPH. The assay was performed in a 96-well microplate and the enzymatic activity was measured following the increase of absorbance at 412 nm using a spectrophotometer (Bio-Rad xMARK^TM^, Microplate Spectrophotometer, Hercules, CA, USA). The activity of glutathione reductase was calculated by the mean of the slopes, obtained at 0 s and at 88 s, using a molar extinction coefficient of 14.15 mM^−1^·cm^−1^, and was expressed as Units/mg.

#### 4.10.3. Superoxide Dismutase Activity

The assay for superoxide dismutase (SOD) activity is based on the reaction in which SOD reduces the superoxide anion to hydrogen peroxide and oxygen. Briefly, a reaction cocktail of pH 7.8 (containing distilled H_2_O; 216 mM phosphate buffer (pH 7.8); 10.7 mM ethylenediaminetetraacetic acid solution (EDTA); 1.1 mM Cytochrome C solution; and 0.108 mM xanthine solution) was incubated for 5 min at 25 °C. Then, we diluted 50 µg of epidydimal protein homogenate in 216 mM of potassium phosphate buffer (pH 7.8) which was added to the reaction cocktail. The reaction was started with the addition of the xanthine oxidase enzyme solution. The assay was performed in a 96-well microplate and the enzymatic activity was measured in a spectrophotometer (Bio-Rad xMARK^TM^, Microplate Spectrophotometer, Hercules, CA, USA) at 550 nm. The activity of the enzyme was calculated by the mean of the slopes, obtained at 0 s and at 88 s. Concentrations of substrates were 0.05 mmol/L for xanthine and 0.025 mmol/L for INT. SOD was calculated by the degree of inhibition and expressed as Units/mg.

#### 4.10.4. Catalase Activity

The catalase activity assay is based on the measurement of the hydrogen peroxide substrate produced by the action of catalase (CAT). Briefly, 50 µg of epidydimal protein homogenate was diluted in the assay buffer (50 mM potassium phosphate buffer, pH 7.0). The reaction was started by the addition of the samples, the assay buffer and 10 mM of H_2_O_2_ solution. This mixture was left to react for 5 min; the reaction was stopped by adding 15 mM sodium azide solution and transferred to a new well. Then, the color reagent (150 mM potassium phosphate buffer, pH 7.0; 0.25 mM 4-aminoantipyrine; 2 mM 3,5-dichloro-2-hydroxybenzenesulfonic acid) was added and the mixture left to stand at room temperature for 15 min. The assay was performed in a 96-well microplate and the enzymatic activity was measured at 520 nm in a spectrophotometer (Bio-Rad xMARK^TM^, Microplate Spectrophotometer, Hercules, CA, USA). The activity of CAT was expressed as activity µmol/min/mL, in which one unit of CAT activity corresponds to the amount of the enzyme that decomposes 1 μmol of H_2_O_2_ in O_2_ and H_2_O per min. at pH 7.0 at 25 °C.

### 4.11. Statistical Analysis

The statistical significance between the experimental groups was assessed by the t-Student test. All experimental data are shown as mean ± SEM (N = 7 for each condition). Statistical analysis was performed using the GraphPad Prism 6 (GraphPad Software, San Diego, CA, USA). Outliers, whenever present, were excluded using the ROUT method, with a q value of 1%. Results were considered significant when *p*-value < 0.05. Due to the low number of animals, which may be a limitation in this study, further analysis of the statistical power (SP) of differences in the experimental data was evaluated with a one-tail test, assuming an alpha of 0.05 that corresponds to a 0.95 confidence interval, as adapted by [55], using the software provided by https://www.sphanalytics.com/statistical-power-calculator-using-average-values/ (accessed on 4 August 2022).

## 5. Conclusions

In sum, T2DM affects the epididymal bioenergetic function by impairment of the molecular axis SIRT1/PGC-1α/SIRT3. The dysfunction of this pathway underlies the control of mitochondrial dysfunction, with compromised antioxidant capacity and increased oxidative stress parameters. An oxidative environment is deleterious to sperm function, compromising male fertility potential. These alterations could explain, in part, the relationship between OS and ? caused by T2DM. However, more studies are required to clarify the mechanisms behind the epididymal bioenergetic function in view of a potential therapeutic target to attenuate the increased decline of male fertility, especially in developed countries where the prevalence of metabolic diseases is a major public health concern.

## Figures and Tables

**Figure 1 ijms-23-08912-f001:**
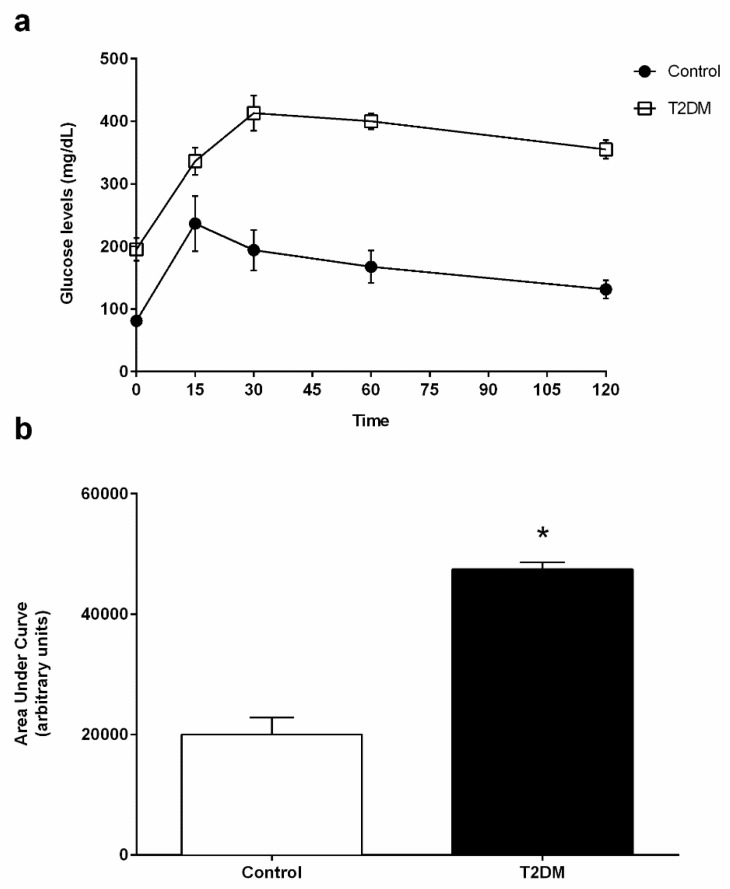
(**a**) Blood glucose levels of control and T2DM groups measured during the intraperitoneal glucose tolerance test (IGTT). (**b**) AUC_GTT_ (area under the curve of glucose tolerance test) analysis of IGTT performed in the control group and in T2DM group. Number of animals per group: 7 in control group and 7 in T2DM group. Results are presented as mean ± SEM. Significantly different results (*p* < 0.05) are indicated: * relative to control.

**Figure 2 ijms-23-08912-f002:**
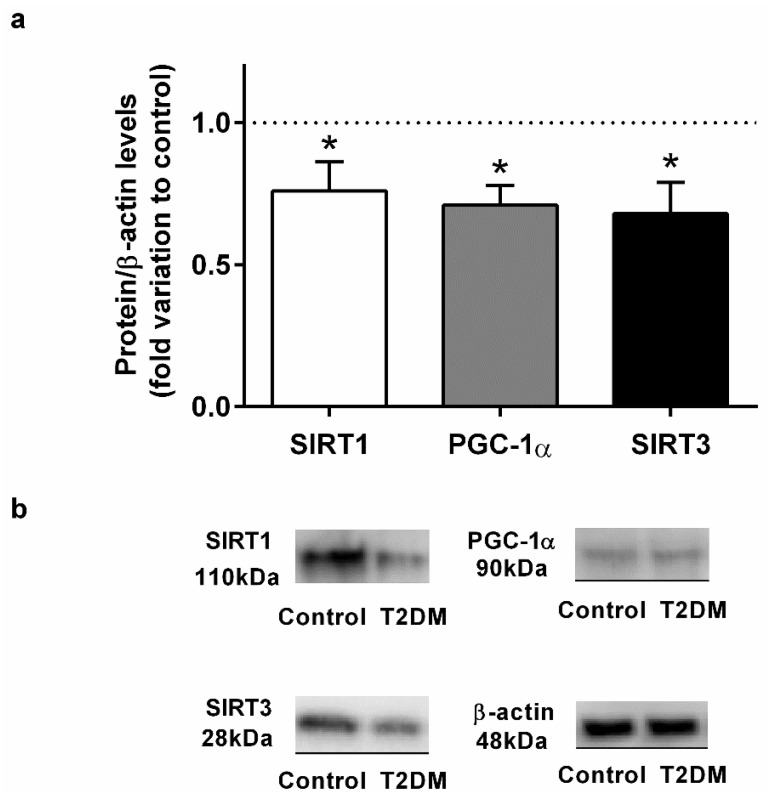
(**a**) Effect of type 2 diabetes mellitus in epididymal proteins levels of Sirtuin 1 (SIRT1), peroxisome proliferator-activated receptor coactivator 1α (PGC-1α), and Sirtuin 3 (SIRT3). (**b**) Representation of an illustrative Western blot experiment for the epididymis tissue of the control and type 2 diabetic animals (T2DM). Number of animals per group: 7 in control group and 7 in T2DM group. Results are presented as mean ± SEM of three independent experiments. Significantly different results (*p* < 0.05) are indicated: * relative to control.

**Figure 3 ijms-23-08912-f003:**
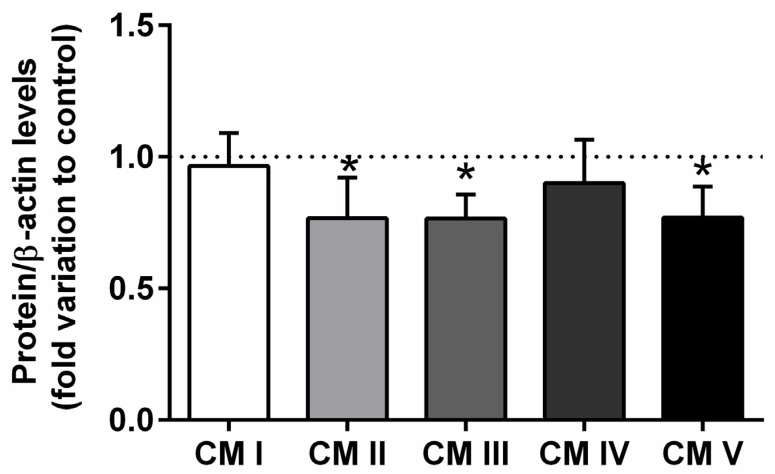
Expression of the mitochondrial complexes I, II, III, IV, V in the epididymal tissue from control and T2DM groups. Number of animals per group: 7 in control group and 7 in T2DM group. Results are presented as mean ± SEM of three independent experiments. Significantly different results (*p* < 0.05) are indicated: * relative to control.

**Figure 4 ijms-23-08912-f004:**
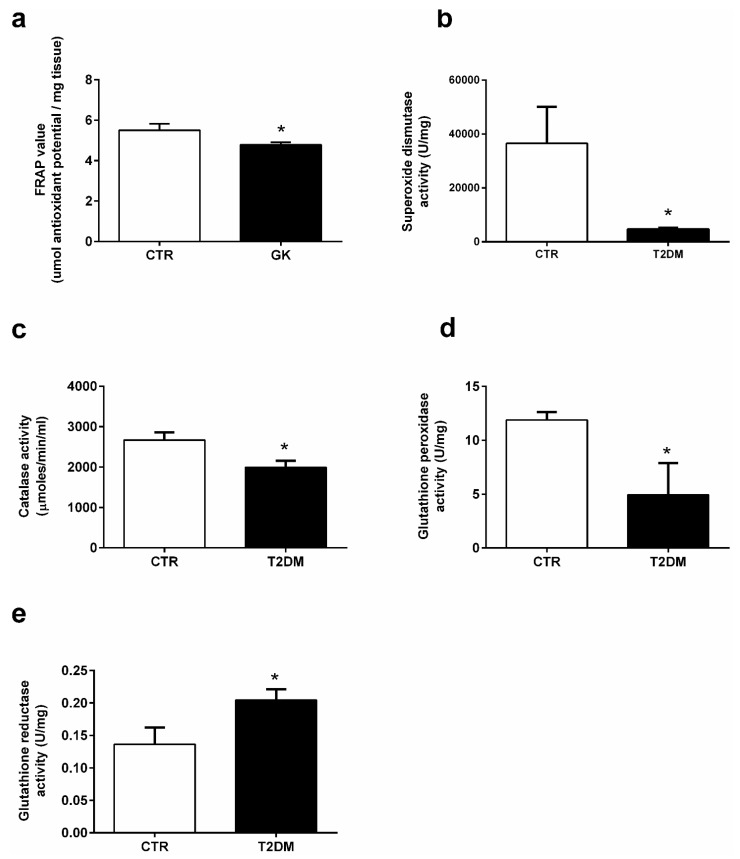
Activities of the antioxidant enzymes (**a**) antioxidant capacity; (**b**) superoxide dismutase, (**c**) catalase, (**d**) glutathione peroxidase, and (**e**) glutathione reductase in the epididymal tissue from the control and T2DM groups. Number of animals per group: 7 in control group and 7 in T2DM group. Results are presented as mean ± SEM of three independent experiments. Significantly different results (*p* < 0.05) are indicated: * relative to control.

**Figure 5 ijms-23-08912-f005:**
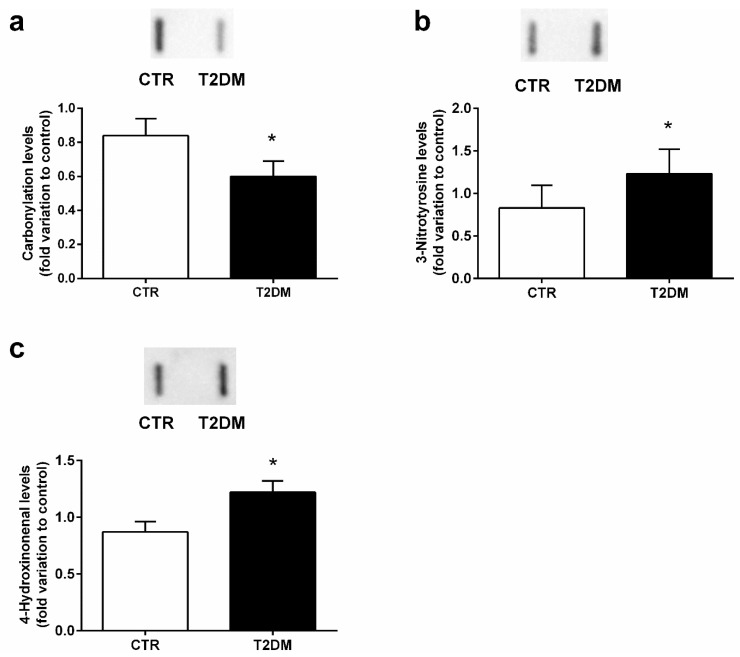
Epididymal antioxidant parameters in the control group and the T2DM group. (**a**) protein carbonylation; (**b**) protein nitration; (**c**) lipid peroxidation. Number of animals per group: 7 in the control group and 7 in the T2DM group. Results are presented as mean ± SEM of three independent experiments. Significantly different results (*p* < 0.05) are indicated: * relative to control. The insets of panel a, b, and c represents an illustrative Slot Blot experiment for the control and type 2 diabetic animals (T2DM).

**Figure 6 ijms-23-08912-f006:**
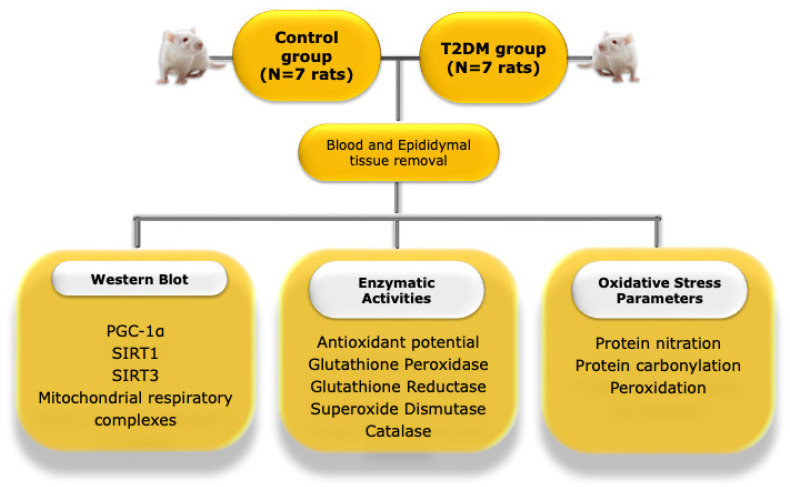
Illustrative representation of the experimental layout of the study. PGC-1α—peroxisome proliferator-activated receptor coactivator 1α; T2DM—type 2 diabetes mellitus; SIRT1—Sirtuin 1, SIRT3—Sirtuin 3.

**Table 1 ijms-23-08912-t001:** Effect of T2DM on average weight, glycemia, HbA1c, insulin, and reproductive organs’ weight from the control group and T2DM group.

Parameters	Control Group	T2DM Group
Weight (g)	470.60 ± 11.95	418.80 ± 6.11 *
Glycaemia (mg/dL)	90.63 ± 6.71	342.50 ± 42.08 *
HbA1c (%)	4.46 ± 0.07	8.29 ± 0.19 *
Fasting plasma insulin levels (ng/mL)	0.15 ± 0.02	0.26 ± 0.02 *
HOMA-IR	0.67 ± 0.07	3.58 ± 0.36 *
Gonads weight (g)	3.77 ± 0.12	2.81 ± 0.03 *
GSI	(0.80 ± 0.03)	(0.66 ± 0.02) *
Epididymis weight (g)	1.47 ± 0.05	1.08 ± 0.02 *
Epididymal fat weight (g)	7.87 ± 0.71	3.99 ± 0.22 *

Legend: GSI: gonadosomatic index; HbA1c: glycosylated haemoglobin; HOMA-IR: Homeostasis assessment model-insulin resistance; T2DM: type 2 diabetes mellitus. Number of animals per group: 7 in control group and 7 in T2DM group). Results are presented as mean ± SEM. Significantly different results (*p* < 0.05) are indicated: * relative to control.

## Data Availability

Not applicable.

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
