# Peer review of "Type 2 Diabetes Induces a Pro-Oxidative Environment in Rat Epididymis by Disrupting SIRT1/PGC-1α/SIRT3 Pathway"

_ijms, 2022, doi:10.3390/ijms23168912_

Round 1
Reviewer 1 Report
In this study, Diniz and colleagues have focused on the involvement of the SIRT1/PGC-1α/SIRT3 Pathway in the evolution of DM2-associated oxidative damage to the male reproductive system. The study is very complex and takes advantage of a good number of molecular techniques to reach the goals set by the study. The results are discussed in a good manner with previous reports. The study reads well. Nevertheless, I would have several questions and/or recommendations:
- The number of animals is quite low. This could be discussed as a potential limitation of the study.
- A number of previous reports have speculated that particularly sirtuins could be modulated by phytotherapy, and in fact there is evidence to indicate that for instance resveratrol could be used as a supporting nutraceutical for the management of metabolic diseases. The authors could add a paragraph on a possible role of biomolecules in assisting in the prevention or management of DM2-associated male subfertility.
- Please add the number of animals in the control/T2DM group into each table/figure footnote.
- Given the complexity of the experimental approach, I would suggest adding a figure or scheme depicting the experimental outline for a higher clarity for the reader.
Author Response
The authors would like to thank this reviewer for the opportunity to improve the work. Below we address all alterations made.
Reviewer #1
In this study, Diniz and colleagues have focused on the involvement of the SIRT1/PGC-1α/SIRT3 Pathway in the evolution of DM2-associated oxidative damage to the male reproductive system. The study is very complex and takes advantage of a good number of molecular techniques to reach the goals set by the study. The results are discussed in a good manner with previous reports. The study reads well. Nevertheless, I would have several questions and/or recommendations:
- The number of animals is quite low. This could be discussed as a potential limitation of the study.
Answer: Following the reviewer’s suggestion, the authors calculated the statistical power of differences in the experimental data evaluated with a one-tail test assuming an alpha of 0.05 that corresponds to a 0.95 confidence interval, as adapted from Levin (2011), using the software provided by https://www.sphanalytics.com/statistical-power-calculator-using-average-values/. Each of the values obtained for the evaluation of the statistical power of differences of the experimental data is in the included corresponding subsections in the Results section.
- A number of previous reports have speculated that particularly sirtuins could be modulated by phytotherapy, and in fact there is evidence to indicate that for instance resveratrol could be used as a supporting nutraceutical for the management of metabolic diseases. The authors could add a paragraph on a possible role of biomolecules in assisting in the prevention or management of DM2-associated male subfertility.
Answer: The authors thank the reviewer’s suggestion and added new information regarding the effects of nutraceutical resveratrol in the management of DM2-associated male subfertility. It can be read from lines 300 to 311.
- Please add the number of animals in the control/T2DM group into each table/figure footnote.
Answer: The authors thank the reviewer and have altered accordingly. It can be read in the table and all figure legends.
- Given the complexity of the experimental approach, I would suggest adding a figure or scheme depicting the experimental outline for a higher clarity for the reader.
Answer: The authors thank the reviewer for this suggestion and agree. A new figure showing all experimental layout was added to the manuscript.

Reviewer 2 Report
The Diniz et al., 2022, Manuscript ID: ijms-1855676 addresses how T2DM induces an antioxidants and Pro-Oxidative environment in the epididymis of Rat by alteration in SIRT1/PGC-1α/SIRT3 pathway. A search on Pubmed.gov for the terms "T2DM" and "Epididymis” and “SIRT1” keywords resulted in very few hits that depicts the novelty of this study. There are few important queries and few suggestion which makes this manuscript more representable to be publish.
1. If it is possible can the author provide histomorphological changes in the epididymis of T2DM rat, so that we can better see the loss of antioxidants?
2. Do the authors have any future plan to do rescue experiments to check the ameliorating effects (such as metformin or any T2DM drug or adiponectin agonist) in the epididymis of T2DM rat?
Author Response
The authors would like to thank the reviewer for the opportunity to revise the work. We kindly ask the reviewer to see the attachment.
Reviewer #2
The Diniz et al., 2022, Manuscript ID: ijms-1855676 addresses how T2DM induces an antioxidants and Pro-Oxidative environment in the epididymis of Rat by alteration in SIRT1/PGC-1α/SIRT3 pathway. A search on Pubmed.gov for the terms "T2DM" and "Epididymis” and “SIRT1” keywords resulted in very few hits that depicts the novelty of this study. There are few important queries and few suggestion which makes this manuscript more representable to be publish.
- If it is possible can the author provide histomorphological changes in the epididymis of T2DM rat, so that we can better see the loss of antioxidants?
Answer: The authors appreciate the reviewer’s suggestion and the possibility to add a picture of epididymis section would provide important information about the effects of T2DM in epididymis. However, let us explain that in this study, the epididymis of each animal was used for: total RNA extraction (not used in this work), protein extraction, intracellular metabolite extraction (not used in this work), and assessment of enzymatic activities. Therefore, due to biological material limitations, the authors decided to focus on the molecular changes of epidydimal metabolism. In fact, the observed molecular changes in this study may infer possible morphological alterations. As a rough measurement of the histological status of the epidydimis, we included in the manuscript the weight of the epididymis of the animals of both groups. Significant differences were observed between the animals of both groups suggesting that the epidydimal architecture was not maintained. Unfortunately, it’s impossible to us add new data concerning histomorphological changes in this tissue. The authors hope the reviewer may understand our point of view.
- Do the authors have any future plan to do rescue experiments to check the ameliorating effects (such as metformin or any T2DM drug or adiponectin agonist) in the epididymis of T2DM rat?
Answer: The authors thank to the reviewer for this important observation. In fact, apart from the animal groups used in this study, two other groups were also stimulated at the same time with Exendin-4 (Ex-4), though we’ve never used this before. Ex-4 is a highly insulinotropic and anti-hyperglycemic agent that shares a 53% amino acid sequence homology with human GLP-1, being resistant to degradation by dipeptidyl peptidase-4. We aim to publish these results first, but we are planning to proceed with the collected material from Ex-4 treated animals to understand the effects of this drug in the reproductive tissues (specifically, testicles and epididymis). We would like to thank you again for this pertinent commentary.
